# Diagnostic accuracy of contemporary and high-sensitivity cardiac troponin assays used in serial testing, versus single-sample testing as a comparator, to triage patients suspected of acute non-ST-segment elevation myocardial infarction: a systematic review protocol

Zhivko Zhelev,[1,2] Hirotaka Ohtake,[3] Mitsunaga Iwata,[3] Teruhiko Terasawa,[3] Morwenna Rogers,[2] Jaime L Peters,[1,2] Chris Hyde[1]

For numbered affiliations see end of article.

**Correspondence to**
Dr Zhivko Zhelev;
Z.Zhelev@exeter.ac.uk

## ABSTRACT

**Introduction** Although the new generation of cardiac troponin assays have revolutionised the diagnosis of myocardial infarction (MI), their application in triaging patients with suspected acute coronary syndrome requires further investigation. The objectives of the current systematic review are to evaluate the diagnostic accuracy of contemporary and high-sensitivity cardiac troponin assays used in serial testing, versus single-sample testing as a comparator, to identify patients with non-ST-segment-elevation MI in the emergency department.

**Methods and analysis** We will conduct systematic searches of MEDLINE, EMBASE, Science Citation Index, the Cochrane Database of Systematic Reviews and the CENTRAL database covering the period from 1 January 2006 to present, with no restrictions on language or publication status. Two review authors will independently screen studies for inclusion, extract data from eligible studies and assess their methodological quality using Quality Assessment of Diagnostic Accuracy Studies version 2. Studies will be included if they evaluate contemporary or high-sensitivity cardiac troponin assays used in serial testing, in patients presenting to the ED with suspicion of MI. Estimates of sensitivity and specificity from each study will be presented in forest plots and in the receiver-operating characteristics space. If appropriate, we will pool the results using Bayesian hierarchical models that allow correction for imperfect reference standard. We will obtain summary estimates of sensitivity and specificity of alternative testing protocols and compare their accuracy. We will also investigate the impact of prespecified sources of heterogeneity and methodological quality items. If pooling of results is considered inappropriate, we will present our findings in tables and diagrams and will describe them narratively.

**Ethics and dissemination** No formal ethical approval will be sought, but we will report on the ethical approval of the included studies. Dissemination of findings will be through publications in peer-reviewed journals, presentations at conferences and the websites of the universities.

### Strengths and limitations of this study

► Systematic and comprehensive searches to identify all published and unpublished studies.
► Robust procedures for screening, data extraction and methodological quality assessment, based on prespecified criteria.
► Statistical methods for pooling results that allow correction for an imperfect reference standard.
► If the number of studies is too small and heterogeneity in study-level estimates is considered too great, we may not be able to obtain summary estimates of sensitivity and specificity.

**PROSPERO registration number** CRD42018106379.

## INTRODUCTION

Coronary heart disease is one of the most common chronic conditions causing more than 7 million deaths worldwide each year.[1] The underlying pathogenesis is progressive accumulation of atheromatous plaque on the walls of the coronary arteries which, in its advanced stages, obstructs the normal blood flow and causes myocardial ischaemia. If the plaque gets ruptured, clots may form and limit, or completely cut-off, the blood supply to part of the heart muscle. This life-threatening condition is known as 'acute coronary syndrome' and presents with chest pain, dyspnoea, nausea and other unspecific symptoms. It includes type 1 myocardial infarction (MI), if there is evidence of permanent damage to the heart muscle, or unstable angina, if no evidence of myocardial necrosis

is detected.[2] MI could also happen in the absence of acute coronary syndrome, as a result of other processes, including mismatch between myocardial oxygen supply and demand (type 2 MI).[3]

Patients with MI require urgent specialist treatment and their outcomes depend on a timely and accurate diagnosis. However, diagnosis in an emergency setting could be challenging, as clinical symptoms lack sufficient sensitivity and specificity[4] and a normal ECG does not rule out the condition. Therefore, in patients suspected of MI but with non-diagnostic ECG, the current guidelines recommend determination of troponin T or I, highly specific biomarkers of cardiac necrosis, both as a triage tool and in establishing the final diagnosis.[5 6]

Although cardiac troponins are specific to myocardial necrosis, they are not specific to MI, and elevated concentration could be found in a number of other acute and chronic conditions that cause myocardial injury. The main difference between acute MI and most of the other conditions is the pattern of dynamic changes. Acute MI is characterised by a sharp increase in the bloodstream concentration of troponin in the first 24 hours of symptom onset, reflecting the progressive nature of the condition. Consequently, a significant difference in troponin concentrations from two consecutive samples (usually referred to as 'delta troponin') is interpreted as an indication that the underlying condition is most likely acute MI. Another consideration is the fact that in the first 2 hours of symptom onset, the level of cardiac troponin in the bloodstream may not exceed the 99th centile, which is an arbitrary cut-off used to dichotomise patients as having or not having myocardial injury. Therefore, a single determination early in the diagnostic process might miss patients with an evolving MI.

### Contemporary and high-sensitivity cardiac troponin assays

Older generations of cardiac troponin assays lacked the precision and sensitivity to identify a small increase in the bloodstream level of the biomarker soon after onset of symptoms. Therefore, testing was done 6–12 hours after presentation to the ED, causing considerable delay in diagnosis and treatment, and unnecessary hospital admissions of patients without MI.[7] In the past 10 years, contemporary and high-sensitivity cardiac troponin assays have been developed and adopted in clinical practice. They can measure small concentrations of troponin with high levels of precision, which means that patients at both high and low risk of having an MI could be identified much earlier in the triage.

Given the large number of cardiac troponin assays marketed as 'high sensitivity', recent efforts have been made to define criteria that could inform policy and clinical decisions. In the development of their guidance on early rule out of acute MI,[5] the UK's National Institute for Health and Care Excellence (NICE) defined a high-sensitivity cardiac troponin assay as one with a coefficient of variation (CV) ≤10% at the 99th centile and an ability to measure concentrations in at least 50% of the reference population. The systematic review that informed the guidance identified only three commercially available assays that met the above criteria: the Elecsys Troponin T high-sensitive assay (Roche Diagnostics), the ARCHITECTSTAT high-sensitive troponin I (Abbott Laboratories) and the AccuTnI+3 (Beckman Coulter).[7] More recently, the Academy of the American Association for Clinical Chemistry and the Task Force on Clinical Applications of Cardiac Bio-Markers of the International Federation of Clinical Chemistry and Laboratory Medicine have endorsed the above definition of high-sensitivity cardiac troponin assay and have recommended 'that assays unable to detect cTn [cardiac troponin] at concentrations at or above the LoD [limit of detection] in at least 50% of healthy men and women be labeled as contemporary cTn [cardiac troponin] assays'. (Wu *et al*, p6)[8]

### Serial testing

Serial measurement of cardiac troponin is employed to increase both sensitivity and specificity of the diagnostic strategy. This usually involves one sample taken at presentation to the ED and another sample taken one or more hours later. The results from these two determinations and their difference ('delta troponin') are combined in a diagnostic rule to determine the likelihood of the symptoms being caused by an acute MI. The strategy takes advantage of the fact that in the first 24 hours of the onset of symptoms, the concentration of cardiac troponin in the bloodstream is progressively rising. According to the current universal definition, diagnosis of MI requires that the patient's presentation is consistent with cardiac ischaemia, cardiac troponin concentration is above the 99th centile of the reference population and two consecutive determinations a few hours apart show a significant change (eg, >20%) in cardiac troponin level.[9] Such patients require urgent reperfusion to restore blood flow to the affected part of the heart and should immediately be admitted to hospital.

### Clinical pathway

Figure 1 shows the clinical pathway for triaging patients suspected of MI, including single-sample and two-sample cardiac troponin algorithms we identified in our scoping searches. Most patients with symptoms suggestive of acute coronary syndrome are transported by ambulance or present directly to the ED.[10] Their initial triage includes clinical assessment, medical history and a 12-lead ECG. Patients with ST-segment elevation MI (STEMI) or other intermediate- or high-risk features are directly admitted to hospital. Low-risk patients undergo troponin testing, with the first sample taken at presentation and the second sample one or more hours later. If both determinations are below a prespecified clinical cut-off, for example, the 99th centile, and there is no evidence of evolving MI (eg, rising troponin, new angina or other symptoms), the patient is diagnosed with unstable angina or alternative explanation of the symptoms is sought.

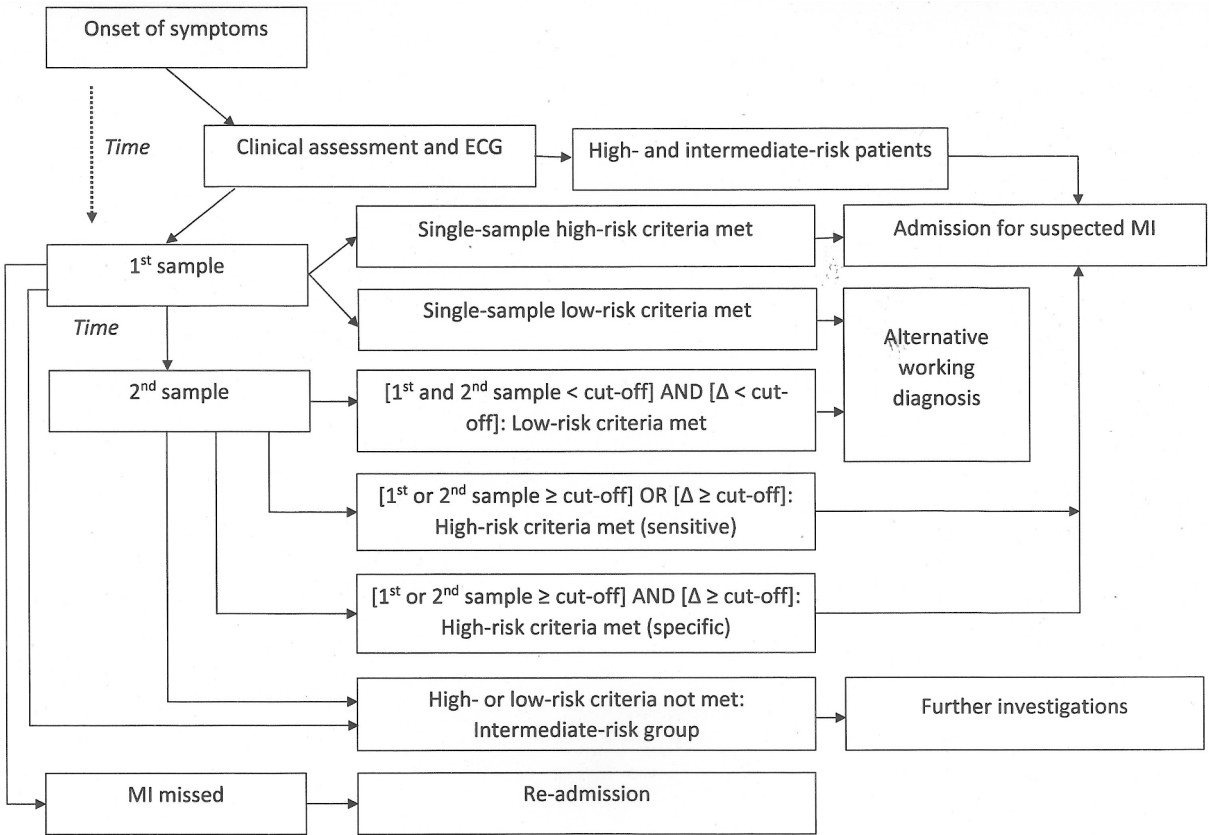

**Figure 1** Clinical pathway for triaging patients suspected of MI, including single-sample and two-sample cardiac troponin algorithms identified in scoping searches. MI, myocardial infarction.

## Diagnostic strategies used in the triage of patients suspected of non-ST-segment elevation MI

Taking advantage of the high sensitivity and precision of the new generation cardiac troponin assays, a range of diagnostic protocols addressing specific clinical scenarios have been devised. They could be summarised as follows:

► Single-sample protocols to identify patients at low risk of MI: If the concentration of troponin in the first sample, taken two or more hours after the onset of symptoms, is well below the 99th centile (eg, below the limit of detection or other prespecified cut-off), the probability of MI is also very low and the patient could be discharged, if appropriate, or alternative explanation of the symptoms could be sought. For instance, a meta-analysis conducted by Chapman and colleagues[11] showed that the negative predictive value of Abbott ARCHITECT$_{STAT}$ high-sensitive cardiac troponin I assay (Abbott Laboratories) is 99.5%, when the cut-off is '<5 ng/L'; the 99th centile of this assay is much higher: 34 ng/L for men and 16 ng/L for women.[12] Alternative early low-risk strategies have also been proposed, for example, combining a single-sample high-sensitivity troponin with copeptin[13 14] or information from the ECG and risk assessment tools, such as the Thrombolysis in Myocardial Infarction risk score.[15 16] This increases the negative predictive value of the protocol and the proportion of patients assigned to the low-risk group.

► Two-sample protocols to identify patients at low risk of MI:
  – If the results from two samples taken one or more hours apart are below a prespecified threshold, such as the 99th centile or lower, and there is no significant change, the probability of MI is low.
  – If both values are slightly above the 99th centile, but there is no change, the underlying cause might be a chronic condition, such as heart failure or myocarditis.

► Single-sample protocols to identify patients at high risk of MI: In the context of myocardial ischaemia, if the concentration of troponin in the first sample is well above the 99th centile, the probability of MI is high and the patient will most likely be admitted to hospital for further testing and treatment.

► Two-sample rule-in: The universal diagnosis of MI requires that at least one of two consecutive samples taken three or more hours apart produces a value above the 99th centile, and there is a significant change in the concentration of cardiac troponin between the first and the second samples.[9]

The above protocols could be combined in complex diagnostic strategies that use different cut-offs to identify patients at low, intermediate or high risk of MI. Such strategies enable clinicians to stratify patients early in the triage, so that appropriate referral and treatment decisions could be made without delay.

## Rationale for conducting the review

Recent meta-analyses have focused on single-sample protocols designed to identify patients at low risk of MI.[11 17 18] Such protocols have important implications for clinical practice, especially in terms of cost-effectiveness and quality of patient care: they could free up capacity required for patients who need emergency treatment; reduce the number of unnecessary tests and the associated risks; the focus of clinical investigation could shift earlier to an alternative explanation of the patient's symptoms and, if appropriate, the patient could be discharged directly from the ED.[10]

However, single-sample protocols are limited in their application, as many patients without MI will present with troponin concentrations above the prespecified cut-off. Such patients will undergo a second troponin determination which, if the inter-sample time is three or more hours, as recommended in the current NICE guidance,[5] will result in the patients spending many hours in hospital. In some cases they will have to stay overnight, until a qualified clinician is available to review results and make a decision for transfer or discharge.

Various serial testing protocols have been devised and validated, including protocols using different cut-offs to identify patients at low and high risk of MI and shorter (1 or 2 hours) inter-sample time. Such protocols could enable discharge or referral decisions to be made for a larger proportion of patients while still in the ED. A brief scoping of the literature showed considerable variation in these protocols and their accuracy, and we can assume that the variation in clinical practice is even greater. The differences concern the following:

► The analytical and diagnostic performance of different assays.
► The cut-offs used to identify patients at low and high risk of MI.
► The method of calculating delta troponin: absolute versus relative change (eg, >4.5 ng/L vs >20%).
► The inter-sample time (one, two or more hours).

Other factors, such as the time from onset of symptoms, patient characteristics (eg, age, sex and risk factors) and prior tests (eg, ECG findings) are also likely to affect the performance of the assays and contribute to the observed heterogeneity in study-level estimates.[11] This could be confusing when making decisions about which protocol to implement in practice in order to achieve the best balance between accuracy and efficiency (the proportion of patients for whom a decision for discharge or admission to a specialist unit could be made early in the triage).

Therefore, we are proposing to systematically identify, review and summarise research evidence pertaining to the diagnostic accuracy of contemporary and high-sensitivity cardiac troponin assays, when used in serial testing to identify MI in symptomatic patients presenting to the ED. Our primary objective will be to obtain summary estimates of the accuracy of specific protocols and the proportion of patients assigned to each risk category. Our secondary objectives will be to determine the relative accuracy of alternative protocols, including a direct comparison of single-sample and two-sample protocols, and to investigate the impact of protocol- and study-level characteristics on test accuracy estimates.

## METHODS AND ANALYSIS

In preparing the protocol, we used the Preferred Reporting Items for Systematic Reviews and Meta-Analyses-Protocol reporting guidelines.[19]

### Eligibility criteria
#### Study design
We will include all primary diagnostic accuracy studies with single-gate design (also known as diagnostic cohort studies). Studies with two-gate design will be excluded due to inherent bias in the reported diagnostic accuracy estimates.[20]

#### Participants
Only studies in patients aged ≥18 years who present to the ED with suspicion of acute coronary syndrome will be considered for inclusion. Studies in patients with STEMI will be excluded, but we will include studies analysing jointly STEMI and non-STEMI patients. We will include data only for non-STEMI, if such data could be obtained; otherwise, we will deal with this in the methodological quality assessment and the sensitivity analysis.

#### Index tests
The index tests are contemporary or high-sensitivity cardiac troponin assays used in serial measurement (two or more determinations) to identify MI. To qualify as 'contemporary' or 'high sensitivity', assays need to have CV ≤10% at the 99th percentile of normal; 'contemporary' assays should be able to detect cardiac troponin in 20% to <50% of healthy individuals and 'high-sensitivity' assays in at least 50%. We will include all cardiac troponin assays that meet the above criteria. We will ascertain eligibility by checking package inserts of the assays, contacting manufacturers and identifying published clinical evaluations.

#### Target conditions and reference standard
Our primary target condition will be an index diagnosis of MI. Our secondary target conditions will be as follow:
► A composite outcome of index MI and cardiac death within 30 days of the index presentation.
► A composite outcome of index MI, death and major adverse cardiac events including cardiac arrest, emergency revascularisation procedure, cardiogenic shock, ventricular arrhythmia or high-degree atrioventricular block requiring intervention, if they occur within 30 days of the index presentation.

The reference standard is a final diagnosis of MI adjudicated by qualified clinicians according to the universal definition of MI.[9] The quality of the reference standard, such as independence from the index test, adjudication

and blinding to the index test results, will be considered in the methodological quality assessment.

## Outcomes

The main outcomes are test accuracy data reported as true positives, false positives, false negatives, and true negatives for protocols with dichotomous outcomes, or three-by-two tables for protocols that assign patients to low-, intermediate- and high-risk categories. If the contingency table is not reported in the paper, we will derive data from the sensitivity and specificity estimates, the total number of patients and the proportion of those with the target condition. Studies that report only test accuracy estimates but insufficient detail to reconstruct the original contingency table will be included in the review, but will be excluded from the quantitative analysis. Studies that meet the rest of the criteria, but do not report test accuracy data, will be excluded only after we have tried but failed to obtain the missing data.

## Publication status and language

We will include conference abstracts and unpublished studies if they meet our eligibility criteria and we are able to obtain the necessary data. We will include all relevant papers regardless of the language in which they are published. If necessary, we will arrange translation or data extraction by a person fluent in the respective language.

## SEARCHES

We plan to perform systematic searches of the following electronic databases: MEDLINE (via OvidSp), EMBASE (via OvidSP), Science Citation Index (via Web of Science), Cochrane Database of Systematic Reviews and CENTRAL database (via the Cochrane Library). We will search all databases from 1 January 2006 to present, with no restrictions on language or publication status. As far as we are aware, no studies evaluating the diagnostic accuracy of high-sensitivity cardiac troponin assays have been published prior to 2010.[7 11 17 18] Hence, the choice of the above publication date as a cut-off for our searches, allowing for some uncertainty in the exact date.

Related systematic reviews will be examined for additional studies for inclusion. Our search strategy complies with the Peer Review of Electronic Search Strategies 2015 Guideline for developing electronic searches. It was developed by an information specialist (MR) with many years of experience of using databases for systematic reviews, and it combines terms for the target condition, index tests and setting. The search strategy for Embase is provided as online supplementary appendix 1 in the Supplemental file, and the search strategies for the other databases could be obtained from the authors on request.

In addition, we will conduct forward and backward searches of all included papers and other relevant publications (eg, previous systematic reviews), and will contact experts in the field and study authors to check for missed titles or unpublished data.

## Selection of studies for inclusion

The results from the electronic searches will be imported into a reference management software and de-duplicated. Two review authors will conduct all screening independently, with disagreements resolved through discussion or arbitration. First, we will screen all titles and abstracts; the full text of any potentially relevant publications will be retrieved and assessed for eligibility against our inclusion criteria. The same procedure will be repeated for titles identified in the additional searches.

## Data extraction

We have developed and piloted a data extraction form available in the online supplement (online supplementary appendix 2 in the supplementary file). Data will be extracted for the following topics: publication details, study characteristics, patients, index tests, reference standard, outcomes and methodological quality items. Two review authors will extract data independently and all disagreements will be resolved through discussion or arbitration. Study authors will be contacted for missing data and to ascertain the independence of study cohorts, with reminders sent 2 weeks after the initial request.

## Methodological quality assessment

We will assess the methodological quality of the included studies using a tailored version of the Quality Assessment of Diagnostic Accuracy Studies version 2 tool. The tool consists of four domains: Patient selection, Index test, Reference standard and Flow and timing. Each domain is assessed for risk of bias and rated as 'High risk', 'Low risk' or 'Unclear risk'. In addition, the first three domains are assessed for applicability concerns and rated using the same categories. The authors of the tool recommend that it is tailored for each systematic review by adding or removing signalling questions.[21] A tailored version of the tool with details of the signalling questions, operational definitions and the rules for combining the answers to produce a domain-level rating is included in the Supplementary file (online supplementary appendix 3).

## Assessment of publication bias

Given the limited knowledge of publication bias in diagnostic accuracy studies and the controversies around its assessment,[22 23] we will not use statistical methods to detect the presence of publication bias. We acknowledge, however, that publication bias might be present and have bearing on our findings. We will try to mitigate this by conducting comprehensive searches of the published research, actively seek to identify unpublished data and note the possibility of publication bias when reporting our findings.

## Statistical analysis and data synthesis

Contingency tables will be reconstructed from the data reported in the papers or by entering sensitivity, specificity, total number of patients and the proportion of those with the target condition in the test accuracy calculator of the Cochrane Collaboration's Review Manager

5.3 (The Nordic Cochrane Centre, The Cochrane Collaboration, 2014). Sensitivity and specificity estimates with 95% CIs will be calculated and presented in forest plots and in the receiver operating characteristics space. Data will be grouped by individual assays and protocols. We will conduct visual inspection of each plot to determine the level and potential sources of heterogeneity and to decide whether meta-analysis is appropriate.

For each assay, if a sufficient number of studies per protocol are available and the level of clinical and statistical between-study heterogeneity is low, we will conduct separate meta-analyses by dichotomising the results from studies reporting more than two outcomes (eg, low-, intermediate- and high-risk categories). First, we will combine the intermediate- and high-risk categories to obtain a two-by-two table for the low-risk rule of the protocol. Then, we will combine the low- and intermediate-risk categories to obtain a two-by-two table for the high-risk rule of the protocol. These will be included in separate meta-analyses to obtain summary sensitivity and specificity for the low-risk and high-risk parts of the protocol.

Since both the index tests and the reference standard are cardiac troponin assays for the detection of myocardial necrosis, we assume the presence of incorporation bias and will pool the results using the model developed by Dendukuri and colleagues.[24] The model, an extension of the hierarchical summary receiver operating characteristic (HSROC) model, allows correction for imperfect reference standard assuming an underlying continuous latent variable. The same methods will also be used to investigate the comparative accuracy of alternative assays and protocols, and the impact of protocol- and study-level characteristics. As a sensitivity analysis, we will also analyse the data using the HSROC model where incorporation bias is not accounted for.

If we decide that pooling of results would be inappropriate (for instance, because the number of studies is too small and heterogeneity is too great), we will summarise the results in tables and graphs, and will provide a narrative summary of the observed differences.

### Investigation of heterogeneity
The protocol and study-level characteristics explored will include the following potential sources of heterogeneity:
► Definition of the target condition: type 1 MI versus type 1 and type 2 MI.
► Different inter-sample time.
► Age (≤65 or >65 years) and sex.
► Time since symptom onset (≤2 or >2 hours).
► Presence of myocardial ischaemia on ECG, for example, ST-segment changes or T-wave inversion.
► History of ischaemic heart disease.

### Sensitivity analysis
If appropriate, we will conduct sensitivity analysis by excluding studies of poor methodological quality and the most influential studies, to assess the robustness of the summary estimates.

### ETHICS AND DISSEMINATION
As this is a systematic review, we are not planning to obtain a formal ethical approval. However, we will extract and report data on the ethical approval of the included studies. The findings from the review will be disseminated through publications in peer-reviewed journals, presentations at conferences and using the websites and networks of the two universities. Changes and amendments to the protocol will be acknowledged in all presentations and publications.

**Author affiliations**
[1]Exeter Test Group, University of Exeter Medical School, University of Exeter, Exeter, Devon, UK
[2]NIHR CLAHRC South West Peninsula, University of Exeter Medical School, Exeter, UK
[3]Department of Emergency and General Internal Medicine, Fujita Health University School of Medicine, Toyoake, Japan

**Contributors** TT originated the idea; TT and ZZ drafted the initial version of the protocol; MR developed the search strategy and wrote the section on search methods; CH, JP, HO and MI reviewed the protocol and suggested amendments. All authors read and approved the final version of the protocol. TT and ZZ are guarantors of the review.

**Funding** This research was funded in part by Fujita Health University, Japan, the Ministry of Education, Culture, Sports, Science and Technology (MEXT), Japan (grant numbers: 18K08902 and 26460755), and the National Institute for Health Research (NIHR) Collaboration for Leadership in Applied Health Research and Care South West Peninsula (NIHR CLAHRC South West Peninsula). The views expressed are those of the author(s) and not necessarily those of the Fujita Health University, the MEXT, the NHS, the NIHR or the Department of Health and Social Care.

**Disclaimer** The funders have no role in the development of the protocol.

**Competing interests** None declared.

**Patient consent for publication** Not required.

**Provenance and peer review** Not commissioned; externally peer reviewed.

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
