## [Reviewer comments · BMJ Open]

ARTICLE DETAILS

TITLE (PROVISIONAL)	Diagnostic accuracy of contemporary and high-sensitivity cardiac troponin assays used in serial testing, versus single-sample testing as a comparator, to triage patients suspected of acute non-ST-segment elevation myocardial infarction (NSTEMI): A systematic review protocol
AUTHORS	Zhelev, Zhivko; Ohtake, Hirotaka; Iwata, Mitsunaga; Terasawa, Teruhiko; Rogers, Morwenna; Peters, Jaime; Hyde, Chris

VERSION 1 - REVIEW

REVIEWER	José LABARERE Grenoble Alpes University School of Medicine and Hospital, TIMC UMR 5525 CNRS, France
REVIEW RETURNED	28-Sep-2018

GENERAL COMMENTS	In their protocol paper, Zhelev et al. report on a planned systematic review with meta-analysis investigating the diagnostic accuracy of cardiac troponin assay for triaging patients with suspected myocardial infarction. The manuscript is clear and well written. The statistical approach is methodologically sound. Yet, I have the following comments: 1. The authors may want to indicate that their review focuses on NSTEMI in the title.2. The title should be consistent with main text regarding the use of serial and single sample testing.3. It may be appropriate to contact the corresponding authors for primary studies with insufficient data to construct the original cross-classification tables.4. Statistical analysis: The authors may want to describe how the direct comparison between single- versus two-sample protocols will be performed?5. Sensitivity analysis: Please provide more details on planned sensitivity analysis. Are the authors planning to perform leave-one out analysis?6. Minor comment: Do the authors comply with the Peer Review of Electronic Search Strategies (PRESS)-2015 Guideline for developing electronic searches?7. Minor comment: Regarding the accuracy of high sensitivity troponin combined with copeptin, the authors may want to cite the following meta-analysis: Raskovalova T et al. Eur Heart J Acute Cardiovas Care 2014;3(1):18-27.
--

REVIEWER	John Pickering Department of Medicine, University of Otago Christchurch, Christchurch, New Zealand
REVIEW RETURNED	08-Oct-2018

GENERAL COMMENTS

This is a well written and clear study protocol. There is much to admire about it, but I have restricted my comments to just some areas that need further consideration. The corresponding author has experience in conducting a meta-analysis in the field previously. I look forward to the results.

Comments:

1. Use “contemporary” rather than “sensitive” for assays that do not reach the standard required to be considered “high-sensitivity”. This is because the International Federation of Clinical Chemistry and Laboratory Medicine have published the recommendation: Recommendation 5: We recommend that assays unable to detect cTn at concentrations at or above the LoD in at least 50% of healthy men and women be labeled as contemporary cTn assays. Reference: Wu AHB, Christenson RH, Greene DN, et al. Clinical Laboratory Practice Recommendations for the Use of Cardiac Troponin in Acute Coronary Syndrome: Expert Opinion from the Academy of the American Association for Clinical Chemistry and the Task Force on Clinical Applications of Cardiac Bio-Markers of the International Federation of Clinical Chemistry and Laboratory Medicine. *Clinical Chemistry*. 2018;clinchem.2017.277186. doi:10.1373/clinchem.2017.277186

Additionally include “contemporary” linked to troponin in the search as without it some articles may be missed.

The authors may like to reference this very recent article as I believe it important for the proposed research.

2. Figure 1: Clinical pathway. Given the vast range of clinical pathways proposed in the field, this is a difficult diagram to produce. However, the boxes that state 1st & 2nd either <99th centile or >~99th centile may be too limiting. While the ESC guidelines 0/3h pathway, the EDACS-ADP, ADAPT-ADP and HEART-ADP are based around the 99th centile, as the authors note many other pathways that are “troponin only” use thresholds well below the 99th centile. I wonder if the figure couldn’t be modified a little to convey what is in the text better.

Additionally the meaning of >~ is unclear. (also I suggest use < and >= rather than < and >).

3. In the discussion on page 12 relating to which cardiac troponin assays meet the criteria described, please state how you are going to determine this (eg via the package inserts of the assays or through a separate systematic search of the literature?).

4. The proposed statistical analysis is good. However, one thing was unclear to me and that is concerning pooling of results. Please forgive me if I appear to be stating the obvious, but I think it needs to be clear in the protocol. For pooling of results, I believe it critical that only studies with identical troponin assays be pooled. The differences between hs-cTnT (Roche) and hs-cTnI (Abbott) are such that it would be inappropriate to pool them (eg there are different proportions of patients below the LoD which is used as a cut-off in the ESC 0/1h protocol). There is evidence also of differences between different hs-cTnI assays.

	Minor comments:  1. Use “concentrations” rather than “values” for measured troponin concentrations (Because concentrations are what are measured, not values). 2. Suggest searching also for pathways explicitly including 0/1h, 0/2h, 0/3h, HEART-ADP, EDACS-ADP, ADAPT-ADP, T-MACS, and the scoring systems HEART, EDACS, TIMI, GRACE. 3. “rule-out”, “rule-in”, “observational zone” are terms that have come from one research group. Others use “Low-risk”, “Intermediate-risk”, and “High-risk.” Notably, some researchers (myself included) have expressed concern that the term “rule-in” erroneously conveys the idea that the algorithm produces a diagnosis when all it does is allocate patients to a risk category for further investigation. Please be cognisant of this and consider how you use the various terms. 4. P10 “positivity threshold” and Appendix 2 “Cut-off valued for positive disease”. I understand that you mean the very low troponin concentration threshold of a particular algorithm, however I expect that many physicians are used to only using the term “positive” for concentrations ≥ 99th percentile. Consider adjusting the language. 5. One of the difficulties in the field is that many studies have been used to develop or validate many different algorithms. I suggest you develop a method to avoid double counting. 6. Appendix 2, Reference standard. I suggest include the assay used to make a diagnosis as this has not always been the same as the assay being tested.
--	--

VERSION 1 – AUTHOR RESPONSE

Reviewer(s) Reports:

Reviewer: 1

Reviewer Name: José LABARERE

Institution and Country: Grenoble Alpes University School of Medicine and Hospital, TIMC UMR 5525 CNRS, France Please state any competing interests or state ‘None declared’: None declared

Please leave your comments for the authors below In their protocol paper, Zhelev et al. report on a planned systematic review with meta-analysis investigating the diagnostic accuracy of cardiac troponin assay for triaging patients with suspected myocardial infraction. The manuscript is clear and well written. The statistical approach is methodologically sound. Yet, I have the following comments:

1. The authors may want to indicate that their review focuses on NSTEMI in the title.

Zhelev & colleagues: We have changed the title as recommended.

2. The title should be consistent with main text regarding the use of serial and single sample testing.

Zhelev & colleagues: We have changed the title as recommended.

3. It may be appropriate to contact the corresponding authors for primary studies with insufficient data to construct the original cross-classification tables.

Zhelev & colleagues: In the Search methods section we state that “...will contact experts in the field to check for missed titles or unpublished data”. To make it clearer, we amended the section by adding “study authors”.

4. Statistical analysis: The authors may want to describe how the direct comparison between single-versus two-sample protocols will be performed?

Zhelev & colleagues: We have already stated that the model developed by Dendukuri and colleagues will be extended to investigate comparative accuracy, provided there is enough homogeneous evidence to allow such analysis.

5. Sensitivity analysis: Please provide more details on planned sensitivity analysis. Are the authors planning to perform leave-one out analysis?

Zhelev & colleagues: We have changed the Sensitivity analysis section stating that sensitivity analysis will include assessing the impact of studies of poor methodological quality and studies identified as most influential in the meta-analysis.

6. Minor comment: Do the authors comply with the Peer Review of Electronic Search Strategies (PRESS)-2015 Guideline for developing electronic searches?

Zhelev & colleagues: Yes. Our database searches were designed and will be ran by an information specialist with many years of experience of using databases for systematic reviews.

7. Minor comment: Regarding the accuracy of high sensitivity troponin combined with copeptin, the authors may want to cite the following meta-analysis: Raskovalova T et al. Eur Heart J Acute Cardiovas Care 2014;3(1):18-27.

Zhelev & colleagues: We have added the reference to the protocol.

Reviewer: 2

Reviewer Name: John Pickering

Institution and Country: Department of Medicine, University of Otago Christchurch, Christchurch, New Zealand Please state any competing interests or state 'None declared': None declared.

Please leave your comments for the authors below This is a well written and clear study protocol. There is much to admire about it, but I have restricted my comments to just some areas that need further consideration. The corresponding author has experience in conducting a meta-analysis in the field previously. I look forward to the results.

Comments:

1. Use "contemporary" rather than "sensitive" for assays that do not reach the standard required to be considered "high-sensitivity". This is because the International Federation of Clinical Chemistry and Laboratory Medicine have published the recommendation:

Recommendation 5: We recommend that assays unable to detect cTn at concentrations at or above the LoD in at least 50% of healthy men and women be labeled as contemporary cTn assays.

Reference: Wu AHB, Christenson RH, Greene DN, et al. Clinical Laboratory Practice Recommendations for the Use of Cardiac Troponin in Acute Coronary Syndrome: Expert Opinion from the Academy of the American Association for Clinical Chemistry and the Task Force on Clinical Applications of Cardiac Bio-Markers of the International Federation of Clinical Chemistry and Laboratory Medicine. Clinical Chemistry. 2018;clinchem.2017.277186. doi:10.1373/clinchem.2017.277186

Zhelev & colleagues: We have updated the manuscript accordingly.

Additionally include “contemporary” linked to troponin in the search as without it some articles may be missed.

Zhelev & colleagues: We have included the term “contemporary” in the updated search strategy.

The authors may like to reference this very recent article as I believe it important for the proposed research.

Zhelev & colleagues: We have added the above publication to References

2. Figure 1: Clinical pathway. Given the vast range of clinical pathways proposed in the field, this is a difficult diagram to produce. However, the boxes that state 1st & 2nd either <99th centile or >~99th centile may be too limiting. While the ESC guidelines 0/3h pathway, the EDACS-ADP, ADAPT-ADP and HEART-ADP are based around the 99th centile, as the authors note many other pathways that are “troponin only” use thresholds well below the 99th centile. I wonder if the figure couldn’t be modified a little to convey what is in the text better.

Zhelev & colleagues: We fully agree and have modified the figure to convey the fact that other cutoffs than the 99th centile are sometimes used as a decision threshold. We also acknowledge in the text that this is not an exhaustive representation of all available algorithms and shows only the ones we found in our scoping searches.

Additionally the meaning of >~ is unclear. (also I suggest use < and >= rather than < and >).

Zhelev & colleagues: We agree and have changed the figure accordingly.

3. In the discussion on page 12 relating to which cardiac troponin assays meet the criteria described, please state how you are going to determine this (eg via the package inserts of the assays or through a separate systematic search of the literature?).

Zhelev & colleagues: We have amended the section stating that we will ascertain eligibility by checking package inserts of the assays, contacting manufacturers and identifying published clinical evaluations.

4. The proposed statistical analysis is good. However, one thing was unclear to me and that is concerning pooling of results. Please forgive me if I appear to be stating the obvious, but I think it needs to be clear in the protocol. For pooling of results, I believe it critical that only studies with identical troponin assays be pooled. The differences between hs-cTnT (Roche) and hs-cTnI (Abbott) are such that it would be inappropriate to pool them (eg there are different proportions of patients below the LoD which is used as a cut-off in the ESC 0/1h protocol). There is evidence also of differences between different hs-cTnI assays.

Zhelev & colleagues: We fully agree and clarified this in the Statistical analysis section.

Minor comments:

1. Use “concentrations” rather than “values” for measured troponin concentrations (Because concentrations are what are measured, not values).

Zhelev & colleagues: We have made changes, as appropriate.

2. Suggest searching also for pathways explicitly including 0/1h, 0/2h, 0/3h, HEART-ADP, EDACS-ADP, ADAPT-ADP, T-MACS, and the scoring systems HEART, EDACS, TIMI, GRACE.

Zhelev & colleagues: We have amended our search strategy accordingly. We did not include “0/1h, 0/2h, 0/3h” as they would generate a significant number of irrelevant hits; all studies that evaluate

such protocols will inevitably include troponin in their title and, therefore, will be captured by the included terms. We have, however, included the other suggested terms, as studies evaluating such protocols may not necessarily included troponin in their titles and keywords.

3. “rule-out”, “rule-in”, “observational zone” are terms that have come from one research group. Others use “Low-risk”, “Intermediate-risk”, and “High-risk.” Notably, some researchers (myself included) have expressed concern that the term “rule-in” erroneously conveys the idea that the algorithm produces a diagnosis when all it does is allocate patients to a risk category for further investigation. Please be cognisant of this and consider how you use the various terms.

Zhelev & colleagues: We agree and replaced all instances with low-, intermediate- and high-risk categories.

4. P10 “positivity threshold” and Appendix 2 “Cut-off valued for positive disease”. I understand that you mean the very low troponin concentration threshold of a particular algorithm, however I expect that many physicians are used to only using the term “positive” for concentrations \geq 99th percentile. Consider adjusting the language.

Zhelev & colleagues: We have made changes using ‘pre-specified cutoff’ to avoid misunderstanding.

5. One of the difficulties in the field is that many studies have been used to develop or validate many different algorithms. I suggest you develop a method to avoid double counting.

Zhelev & colleagues: We added a sentence in the Data extraction section to state that study authors will be contacted to ascertain independence of data sets.

6. Appendix 2, Reference standard. I suggest include the assay used to make a diagnosis as this has not always been the same as the assay being tested.

Zhelev & colleagues: We have amended Appendix 2 as recommended.

VERSION 2 – REVIEW

REVIEWER	John Pickering University of Otago Christchurch I have been involved in assessing or implementing several of the pathways and troponin assays that the authors may identify in their search.
REVIEW RETURNED	03-Feb-2019

GENERAL COMMENTS	My initial comments have been adequately dealt with. I wish the authors all the best with their proposed research. I look forward to the results.
---